# OpenReview forum: "Report Cards: Qualitative Evaluation of Language Models Using Natural Language Summaries"
_ICLR.cc/2025/Conference — Submitted to ICLR 2025_

### Official Review · Reviewer_D9R8 · 2024-11-02

**Soundness:** 2
**Presentation:** 4
**Contribution:** 2
**Rating:** 5
**Confidence:** 4

**Summary:**

The paper introduces Report Cards, a technique to evaluate LLMs using natural language instructions, and provide interpretable and holistic results.
The authors present the algorithm for progressively generating report cards, PRESS - it features accumulating / summarizing LLM judgement results of models' predictions for text-based tasks. They as well present 3 ways to evaluate Report Cards themselves: the Contrastive Accuracy (given 2 model predictions and 2 report cards, a judge is asked to match a report card to the prediction), Card Elo (given 2 predictions and 2 cards, determine a winner), and human scoring.

They evaluate Report cards on 3 datasets (MMLU, Adv. AI Risk and CN Grammar) using a range of small to large backbone models.
In the contrastive learning judgement, they compare Report Cards with 2 baselines: Constant (judging based on accuracy on ground truth labels) and few-shot (instead of report cards, the contrastive judge is given k predictions the training set)
Card Elo is evaluated via R2 score against Ground Truth Elo (for which either gold labels or another LLM judge are used). Finally, in human scoring, the cards were assessed on a 5-point Likert scale along Relevance, Informativeness and Clarity dimensions.

Evaluations show that Report cards achieve competitive results using the proposed Contrastive Evaluation technique, although with an unexpectedly high result of the few-shot baseline; Card Elo achieves superior R2 scores against competing Elo methods and baselines; Report Cards also attain consistently high scores in human evaluation.

**Strengths:**

* a new method for evaluation of models on text generation tasks is proposed
* it provides high human interpretability and achieves high accuracy together with 2 evaluation methods, Contrastive Evaluation and Card Elo

**Weaknesses:**

* few-shot baseline for Contrastive evaluation is confusing - it's unclear to me how informative is matching a model's prediction on testset to k sampled predictions on the trainset
* de-stylization issue suggests that few-shot baseline captures not very useful information and should ideally be re-thought
* Ground truth Elo for R2 faithfulness evaluation uses yet another judge - so it should probably be renamed into Oracle Elo

**Questions:**

N/A

---

> ### Author Response · Authors · 2024-11-24
> **Rebuttal by Authors**
>
> Thank you for your valuable review and suggestions. We appreciate your detailed and informative concerns. Below, we address the comments on **Weaknesses (W)**.
>
> > Note: We've marked "D9R8" next to relevant changes in the general comments to make it easier for you to navigate.
>
> **W1. Clarification re: few-shot baseline**
>
> The contrastive task measures specificity - how well a qualitative evaluation captures model-specific characteristics. When humans compare models, they naturally examine a few examples to identify consistent behavioral traits that could predict performance on new tasks. Our few-shot baseline implements this intuitive process: using k training examples per model to simulate how humans derive and apply their understanding of model behavior. Since Report Cards are also generated using the training set, this baseline provides a fair comparison to evaluate whether our structured summaries can capture model-specific traits as effectively as the natural example-based approach.
>
> **W2. De-stylization results suggests few-shot baseline captures stylistic information**
>
> We appreciate the reviewer's insight about the few-shot baseline's limitations. We agree that the de-stylization results reveal that few-shot examples primarily capture superficial stylistic patterns rather than substantive behavioral traits. **However, this finding actually strengthens our evaluation: Report Cards maintain strong performance even after de-stylization**, particularly on Advanced AI Risk topics where the few-shot baseline's high performance was driven by exploiting stylistic cues. Together, the few-shot baseline and de-stylization experiments demonstrate that Report Cards successfully capture meaningful behavioral characteristics rather than superficial patterns. In addition, the one-pass generation method also servers as a baseline (Figure 8), supporting the performance of PRESS.
>
> **W3. Rename to Oracle ELO**
>
> We agree, and we will rename it to Oracle Elo. Thanks for pointing that out!

---

> > ### Comment · Reviewer_D9R8 · 2024-11-24
> > **Follow-up**
> >
> > Thanks to the authors for providing clarifications on the questions I pinpointed. Now I have a follow-up question: if both report cards and few-shot examples for the baseline were generated from the trainset, do I understand it correctly that the evaluation took place on the trainsets of the datasets you used? And if so, wouldn't that make it harder to compare this method to some other methods that actually tune on the train/dev sets?
> > If I got it all wrong, could you please clarify how exactly you use train/dev/test sets in your evaluation?

---

> ### Author Response · Authors · 2024-11-24
> **Clarification on train/test split**
>
> All results (contrastive/faithfulness metrics) are reported based on held-out test sets that have no overlap with the training sets used to create the Report Cards and/or select samples for the few shot baselines.
> In one out of all of our datasets (*MMLU High school mathematics*), we did use the test set as a validation set in order to tune prompt. Otherwise, the test sets have been held out.
>  See Page 15 for details (the train set is 40 examples and test set is 60 examples). You mention other methods to compare against, is there something specific you have in mind?
>
> We hope this answers your question and are happy to clarify further.

---

### Official Review · Reviewer_BghS · 2024-11-03

**Soundness:** 2
**Presentation:** 3
**Contribution:** 2
**Rating:** 5
**Confidence:** 4

**Summary:**

The paper suggests a qualitative way to summarize an LLM's performance on a benchmark using "Report Cards". E.g., for Llama-70B-Instruct on MMLU High School Physics, the report card describes strength/weaknesses across ~10 dynamic facets, like "* Newton's Laws Mastery: The student demonstrates a solid understanding of Newton's laws, ...  However, it shows a misunderstanding of Newton's third law ...".

They generate report cards for 9 models across 10 tasks, evaluating their quality through some proxy metrics: 1) Contrastive accuracy: How well can a "guesser LLM" look at two models' reponses to a quiz (k questions in a task) along with a report card for each model, and guess which responses are from which model. 2) Card Elo: A "judge LLM" compares to report cards to decide which is the "best" model, to derive a "Card Elo" rating. How well does that agree with some "ground truth" Elo rating of the models? 3) Human scoring of report cards clarity, relevance, informativeness.

They use an "evaluator LLM" to generate the report cards, trying two main approaches (one-pass prompting and iterative PRESS method). The results on the proxy metrics indicate that the iterative method works better, and also is better than some other baseline approaches for each metric.

**Strengths:**

The paper's focus on getting more insights into LLM performance, beyond just accuracy numbers, is an important topic. The presentation is reasonably clear, with good descriptions of the approach.

**Weaknesses:**

The utility of these report cards is very unclear, the few concrete examples given in the paper give little confidence that any actionable insights can be derived.

The report cards are trying to identify useful subtopics for a task, and describe a model's performance on such sub topic. A much more concrete way of doing that would be to generate some hypotheses (like the example in Figure 10 that Llama-3-8B-Instruct "struggles with combinatorial concepts".) based on seeing a subset of the dataset. Then associate that with a score on that subtopic (model scores 45% on questions with "combinatorial concepts" vs 65% overall). Then further test that hypothesis on a held out set of questions to see if the drop in performance hold up. This seems like a minimal expectation for this kind of work.

The proxy metric of "Contrastive accuracy" seems like it would have many confounders, and a strong dependency on the size of the quizzes (which sounds like it's 120 questions, it's a bit unclear). Some attempts are done to "de-stylizing" the model outputs, which deals with the confounder of CoT style, that's a good idea. But as a measure of report card utility, it seems quite farfetched.

The Card Elo metric is extracting a measure of overall performance from the report card itself (according to a judge LLM), computing correlation with Elo based on actual task performance. They get reasonable R^2 scores of around 0.8, beating out R^2 from a generic ChatBot Arena Elo. It's far from perfect in ranking the models though, so not a substitute for the raw score.

This section has a quote "Importantly, k-shot completion Elo, using the same number of comparisons as Card Elo, obtains a significantly worse faithfulness score than Card Elo." which is a bit unclear in what is meant. It looks like "k" in this context is 2 or 4 (according to section 3.2, this is a different "k" from the 120 apparently used for quiz sizes?). So unclear what "same number of comparisons" means here, maybe the number of model matchups which seems a bit irrelevant (there's only a fixed number of models to possibly match up here).

Finally, the human evaluation (and associated LLM-simulating-human evaluation) of the report cards could be the most insightful. But the most interesting "informativeness" score (which averages a high 4.5 out of 5 from the humans) is based on a generic question: "How informative is the excerpt about the model's capabilities with respect to the question and the model answer?" with choices like

4. Very informative: Provides comprehensive information, covering most key aspects.
5. Extremely informative: Provides extensive, detailed information, covering all key aspects.

which tells us very little about how effective this report card is to actually understand something useful about the models' performance.

Admittedly this is a hard task to evaluate, but the current effort leaves me with no confidence that these report cards are actually useful for anything.

Also missing is any analysis across the models' report cards to hypothesize actionable differences, both for improving the models, or at least to estimate performance on new subsets.

**Questions:**

NA

---

> ### Author Response · Authors · 2024-11-24
> **Rebuttal by Authors**
>
> Thank you for your valuable review, and criticisms. We appreciate your detailed and informative concerns. Below, we address the comments on **Weaknesses (W)**.
>
> > Note: We've marked "BghS" next to relevant changes in the general comments to make it easier for you to navigate.
>
> **W0. The utility of these report cards is very unclear, the few concrete examples given in the paper give little confidence that any actionable insights can be derived.**
>
> While we agree that our paper does not provide many examples of actionable insights, this was not our stated intent, and we respectfully disagree with the reviewer’s characterization of the utility as “unclear”. As noted in our work, quantitative metrics such as MMLU accuracy provide limited insight into underlying capabilities, and some form of accompanying, interpretable explanation is necessary for humans to understand the causes of the quantitative performance. We propose Report Cards, as they possess several advantages over sample-based explanations (e.g., the few shot baseline in our work). Our experiments are not intended to prove actionable insights (the existence of which is independent of the quality of the qualitative evaluation), but rather to show specificity, faithfulness, and interpretability, which we argue is more important for purposes of evaluating qualitative evaluations (in particular, we disagree that immediately actionable insights are necessary for utility; see e.g., the GPT-4 System Card or the Sparks of AGI Paper — understanding behavior is, in and of itself, a utility).
>
> We elaborate on your more specific criticisms below.
>
> **W1. A much more concrete way of [trying to identify useful subtopics for a task, and describe a model's performance on such sub topic] …**
>
> If we understand correctly, you are proposing that given a dataset, we identify a subset $T$ of questions that pertain to some semantically meaningful topic (e.g., combinatorial reasoning) and then see whether a quantitative metrics computed on the training subset $T_{\text{train}}$ generalizes to the test subset $T_{\text{test}}$. However, the $T_{\text{train}}$ performance will always generalize to $T_{\text{test}}$ long as the method of defining subset $T$ is the same for both the train set and test set, so we believe this suggestion would not prove anything about the quality of Report Cards.
>
> While your method would allow us to obtain model-specific accuracy figures for certain subsets, which naturally meets our “specificity” requirement, this still leaves open the question of whether the subset identified is faithful to its natural language description.
>
> If we have misinterpreted the suggestion, we would appreciate your clarification.
>
> **W2 “Contrastive Accuracy” has** **confounders; confusion of $k$ in the few-shot baseline**
>
> Besides stylistic features, which we address in our experiment, could you please identify the confounders you are concerned with? Regarding the confusion in the size of quizzes $k$:
>
> - In the contrastive task, the guesser LLM receives two Report Cards (one for each student LLM) and two sets of responses (one from each student in random order) to a quiz consisting of $k=3$ questions. For each topic and student pair (72 pairs in our paper), we perform this task 120 times. As we mentioned in the paper, the accuracy for each topic is averaged over 8,640 samples for each topic, and the variance is negligible.
> - We’ve revised the few-shot baseline description. The few-shot baseline involves $l$ examples of model responses. These $l$-shots are then used in place of Report Cards, as a baseline, in both the contrastive task and the Elo task. We apologize for the confusion.
>
> **W3. As a measure of report card utility, [contrastive accuracy] seems quite farfetched**
>
> As noted above, the contrastive metric is not a measure of the utility of Report Cards, but rather a measure of the specificity of qualitative evaluations. The specificity is defined as the nuances specific to the model, and by contrasting and matching Report Cards with correct responses, we can show how good our Report Cards are capturing such nuances.
>
> **W4. Card Elo is not a substitute for the raw scores.**
>
> Similarly, the Card Elo task is a measure of the faithfulness - how good do qualitative evaluations capture the true model capability - instead of a substitute of the raw score. By comparing with ChatBot Arena Elo, we are showing that Report Cards are better in capturing true capabilities of models on a fine-grained level. We are not trying to rank models based on Report Cards on a dataset/benchmark where quantitative scores are available.

---

> > ### Comment · Reviewer_BghS · 2024-11-26
> > **Response to authors**
> >
> > Thank you for your comprehensive responses, and especially for the new supplementary material which is very helpful.
> >
> > The report_cards.html comparison tool is great for comparing models. I think the numbers were supposed to show example questions for each part of the report? Although for me they're always showing some math questions, regardless of the topic, so probably a small bug there? I do think showing actual examples along with the general analysis can be quite useful in understanding model behavior.
> >
> > I have raised my scores a bit, although after having read through several more of these report cards, I am still left wondering "what am I really learning about these models?" and "what will I do with that information?". I wonder if there is a version of these report cards which focuses more on "surprises" of sorts, like unexpected strengths/weaknesses compared to what you might expect from the general level (overall accuracy) of the student, or something like that.
> >
> > Anyway, I agree that qualitative evaluations can be interesting in their own right, but they ideally would tell some sort of story. In the "sparks of AGI" paper, that story was about never-before-seen types of behaviors that were interesting to see examples of.

---

> > > ### Author Response · Authors · 2024-12-01
> > > **Prompt Bug Fix**
> > >
> > > We sincerely appreciate your detailed feedback and your comments on qualitative evaluations. Apologize for the bug - we will fix it promptly.

---

### Official Review · Reviewer_f1vs · 2024-11-04

**Soundness:** 3
**Presentation:** 4
**Contribution:** 3
**Rating:** 8
**Confidence:** 4

**Summary:**

This work proposes Report Cards, human-interpretable natural language summaries, based on three criteria: specificity, faithfulness, and interpretability. They also propose PRESS an iterative algorithm for generating report cards. They show through experimentation with popular LLMs that report cards provide insights beyond traditional benchmarks.

PRESS (Progressive Refinement for Effective Skill Summarization) is an iterative algorithm for generating Report Cards without human supervision. PRESS works by iteratively generating and refining summaries of model behavior, focusing on specific aspects in each iteration and synthesizing them into a comprehensive overview.

The authors conducted experiments across multiple datasets, including MMLU (Massive Multitask Language Understanding), the Anthropic Advanced AI Risk dataset, and a Chinese grammar dataset. They tested various models, from OSS models like Llama-3.1-8B-Instruct and Mistral-7 B to closed models like GPT-4 and Claude 3.5 Sonnet.

**Strengths:**

1) There is a need for more than model cards in the LLM information space, while model cards report model parameters, and other minute details like data, training steps (including post training), and evaluations on various benchmarks, they do not provide holistic insights into model capabilities and nor are these insights granular in nature. Evaluations on benchmarks are ultimately unidimensional and limited to the number, and do not provide specific insights into what the model is good or bad at.

2) It also perhaps addresses a real need in LLM development and deployment, where model providers or users are interested in models that are better at specific tasks (or sub tasks) vs overly general models.

**Weaknesses:**

1) The PRESS algorithm ultimately depends on a judge model (in this case, Claude) to provide the final report card; usage of Claude or any model at that scale would add some costs to the generation of the report card vs. model cards that do not rely on LLM judgement.

2) The reliance on LLM as a judge model perhaps also showcases a shortcoming, would a model be good at verifying or recognizing a model better than it or rely on its own (perhaps wrong) judgement.

3) It is not wholly clear if the report card works for advanced topics, MMLU while it covers a variety of topics, is not a benchmark covering all the depth and breadth of any one subject.

**Questions:**

1)) Assuming a specific case, a question that cannot be reliably solved by the judge model as well, but a fine-tuning of the base model does solve it, can a judge model reliably generate a summary for the same? Is the method generally constrained by the capability of the larger, better model? Is there any recourse for this?
2)  Is the readability of the report card considered at the macro level? What is done to balance specificity vs general comments? Figure 10 is a very good example of capturing nuances. Still, if this is done regularly at a global level across questions and subjects, this would quickly add to the length of the report card, and thus, the readability of the report card would be affected, right?

Could you showcase a full report card for any model of your choice? While the methodology to create it makes sense, it is not quite the same as looking at the final product.

---

> ### Author Response · Authors · 2024-11-24
> **Rebuttal by Authors [1/3]**
>
> Thank you for your valuable review, concerns, and questions. We appreciate that you recognize our work as potentially addressing the need for “more than model cards”. Below, we address the comments on **Weaknesses (W)** and **Questions (Q).**
>
> > Note: We've marked "f1vs" next to relevant changes in the general comments to make it easier for you to navigate.
>
> **W1. Use LLMs as Report Card writers at a scale add some cost.** We used `Claude 3.5 Sonnet` as the *card writer* model. In the current PRESS algorithm, it would cost ~$0.1 to generate a Report Card, which is not too expensive. Notice that only one card is needed for each model under a topic, so in actual use cases when users want to qualitatively check the performance of models on a specific task (topic), the total number of cards would equal to the number of models. The costly tasks are the metrics - assessments of Report Cards themselves - which are not required for typical use cases.
>
> In addition, we have conducted an ablation study on different card writer models with varying capabilities and costs.  Using the same experiment setup as the ablation studies in our paper (Appendix Section C), we report the specificity and faithfulness on the MMLU dataset. The results are aggregated over topics as in Appendix Table 6.
>
> | Card Writer       | Specificity | Faithfulness | Cost per 1M Token (input, output) |
> | ----------------- | ----------- | ------------ | --------------------------------- |
> | Claude 3.5 Sonnet | 0.71        | 0.88         | 3, 15                             |
> | Claude 3.5 Haiku  | **0.71**    | **0.86**     | 1, 5                              |
> | Claude 3 Haiku    | 0.61        | 0.50         | 1, 5                              |
> | GPT-4o            | 0.72        | 0.87         | 2.5, 10                           |
> | GPT-4o-mini       | **0.70**    | **0.80**     | 0.15, 0.6                         |
> | GPT-3.5-turbo     | 0.60        | 0.80         | 0.5, 1.5                          |
> | Llama 3.1 70B     | 0.67        | 0.79         | N/A                               |
>
> As the table shows, smaller and cheaper models like Claude 3.5 Haiku, Llama 3.1 70B, and GPT-4o-mini can generate reasonably good Report Cards at a lower cost (it costs only $0.005 to generate a Report Card with GPT-4o-mini). We also see that Claude 3.5 Haiku produces Report Cards of the same quality as Claude 3.5 Sonnet while being more cost-effective.

---

> ### Author Response · Authors · 2024-11-24
> **Rebuttal by Authors [2/3]**
>
> **W2: The reliance on LLM as a judge model perhaps also showcases a shortcoming, would a model be good at verifying or recognizing a model better than it or rely on its own (perhaps wrong) judgement.**
>
> To address the concern about whether a card writer might rely on its own potentially incorrect judgments when evaluating a more capable student model, we conducted an ablation study. By comparing Report Cards generated with and without access to ground truth answers, we can determine if the card writer model's assessments are overly dependent on its own understanding rather than objective evaluation criteria. To test this, we specifically selected `claude-3-haiku`, a model that consistently performs poorly on the MATH dataset, as our card writer model. The ablation study is conducted with the same setup as the card writer ablation, but on the MATH dataset. The results of this investigation were:
>
> | MATH Topic               | `claude-3-haiku` accuracy | Faithfulness with Ground Truth | Faithfulness w/o Ground Truth |
> | ------------------------ | ------------------------- | ------------------------------ | ----------------------------- |
> | algebra                  | 0.29                      | 0.94                           | 0.40                          |
> | prealgebra               | 0.3                       | 0.70                           | 0.63                          |
> | number theory            | 0.15                      | 0.56                           | -0.12                         |
> | counting and probability | 0.08                      | 0.48                           | 0.22                          |
>
> This indeed shows that `claude-3-haiku` is relying on its own errored judgements when generating Report Cards. However, we also show that a more intelligent model with higher dataset accuracy, but far from good, would achieve *consistently high faithfulness*.
>
> | MATH Topic               | `gpt-4o-mini` accuracy | Faithfulness with Ground Truth | Faithfulness w/o Ground Truth |
> | ------------------------ | ---------------------- | ------------------------------ | ----------------------------- |
> | algebra                  | 0.74                   | 0.91                           | 0.97                          |
> | prealgebra               | 0.60                   | 0.66                           | 0.87                          |
> | number theory            | 0.67                   | 0.87                           | 0.82                          |
> | counting and probability | 0.45                   | 0.82                           | 0.71                          |
>
> In conclusion, the quality of Report Cards relies moderately on card writer capability, but (1) we can mitigate the issue by including ground truths and (2) Report Cards generated by a reasonably performed card writer have high faithfulness.
>
> **W3: It is not wholly clear if the report card works for advanced topics, MMLU while it covers a variety of topics, is not a benchmark covering all the depth and breadth of any one subject.**
>
> To show that Report Cards works for advanced topics, we sample from the **most difficult problems** in the MATH dataset (level 5).
>
> Note that due to computational constraints during the brief rebuttal window, the results below use `gpt-4o-mini` as the guesser to assess the specificity and the faithfulness instead of `Llama-3.1-405B` (as in our paper), while `gpt-4o-mini` is a good indicator, the accuracies would be higher with a stronger guesser. To maintain methodological consistency in our paper, we decided not to include these results in the current version, as they were obtained using a different guesser than our main experiments. We plan to replicate these experiments using `Llama-3.1-405B` when we have access to adequate computational resources.
>
> | MATH Topic               | Claude 3.5 Sonnet Accuracy | RC Specificity | Few-Shot Specificity | RC Faithfulness | Few-Shot Faithfulness |
> | ------------------------ | -------------------------- | -------------- | -------------------- | --------------- | --------------------- |
> | algebra                  | 0.75                       | **0.58**       | 0.51                 | **0.96**        | 0.66                  |
> | prealgebra               | 0.75                       | **0.64**       | 0.52                 | **0.89**        | 0.32                  |
> | number theory            | 0.61                       | **0.60**       | 0.49                 | **0.80**        | 0.73                  |
> | counting and probability | 0.47                       | **0.60**       | 0.52                 | **0.70**        | 0.31                  |

---

> ### Author Response · Authors · 2024-11-24
> **Rebuttal by Authors [3/3]**
>
> **Q1: Assuming a specific case, a question that cannot be reliably solved by the judge model as well, but a fine-tuning of the base model does solve it, can a judge model reliably generate a summary for the same?**
>
> As we've shown in response to W2, we can generate Report Cards with high faithfulness by using reasonably performing models (like in the W2 case, `gpt-4o-mini`). We can also include the ground truth answers (if available) when generating Report Cards to mitigate the issue. In cases where there are no ground truth answers, Report Cards can be easily augmented with few-shot samples to show both the interpretable and condensed qualitative evaluations (Report Cards), and the actual examples of model behaviors. We've attached a HTML file `report_cards.html` to the supplementary materials for you to access the augmented Report Cards across many topics and models.
>
> **Q2: Is the method generally constrained by the capability of the larger, better model?**
>
> As noted in the response to weakness 1, smaller and less-capable card writer model like `gpt-4o-mini` and `claude-3.5-haiku` can generate Report Cards with similar level of specificity and faithfulness. However, with stronger card writers, the Report Cards can indeed capture more nuances and faithful behaviors, and we expect that Report Cards will improve as our models improve.
>
> **Q3: Is the readability of the report card considered at the macro level? What is done to balance specificity vs general comments?**
>
> Indeed. Readability at a global level is impacted by our focus on nuanced behaviors. However, each bullet point in the Report Card focuses on independent subtopics or criteria. During human evaluations, a LLM is to extract relevant parts from the whole Report Card. We anticipate that for general usage, a practitioner can get the relevant excerpt (such as via search or LLM extraction) instead of the longer complete card.
>
> In addition, we also proposed the Hierarchical Bullet Point Format for Report Cards (Appendix Section A), which captures the behaviors with both high level descriptions (`overview` and `thinking_pattern`) and fine-grained details (`strength` and `weakness`). HBP performs slightly worse than the bullet point format as shown by Figure 12 (d), so we chose the Bullet Point format as our main format.
>
> **Q4: Could you showcase a full report card for any model of your choice?**
>
> We have attached some example complete Report Card with many augmented Report Cards examples. Please check the general comments for details. Thanks!

---

> > ### Comment · Reviewer_f1vs · 2024-11-26
> > **Response to Authors Rebuttal**
> >
> > Thank you for your responses and the additional experiments. I am satisfied with the answers and am increasing my score to 8 with a confidence of 4.
> >
> > I would like to suggest that the paper clearly state "specificity vs general comments" regarding subject coverage and the constraint of "better models for better cards." This would improve the work and also make readers aware of the limitations. While the idea of using ground truth to empower smaller models to generate report cards sounds good, it would not be feasible for advanced content, and collecting this ground truth itself could also add additional costs.

---

> > > ### Author Response · Authors · 2024-12-01
> > > **Inclusion of Subject Coverage**
> > >
> > > We sincerely appreciate your careful consideration and positive assessment of our work. We will include the subject coverage (additional MATH dataset difficulty questions) and the constraint investigation (also with MATH) in our paper once we obtain the results with a consistent guesser as in the paper. Unfortunately, given our resource constraints, it is not feasible to rerun them within the discussion time window.

---

### Official Review · Reviewer_EuP9 · 2024-11-04

**Soundness:** 3
**Presentation:** 3
**Contribution:** 3
**Rating:** 5
**Confidence:** 4

**Summary:**

This paper proposes a framework for qualitative evaluation, based on specificity, faithfulness, and interoperability. To do so, the paper proposes a PRESS algorithm to generate natural language descriptions for LLMs in an iterative way. The experiments cover multiple LLMs, and the corresponding analyses read well.

**update after rebuttal**: I am not fully convinced by the authors' response, in particular the human evaluation part. If the authors suggest that they need more time to collect data for reliable evaluation, I suggest they fully prepare it and submit the paper to other suitable conferences.

**Strengths:**

This paper is generally well-written and is very interesting.

**Weaknesses:**

1. There are other works focusing on qualitative evaluation, such as [1], which formulate the problem in a very similar way, i.e., generating human languages to describe the limitation of LLMs.
2. There are some unavoidable and major limitations, although the authors have discussed them in the limitation section. For example, the selection of tasks is limited, and seems that the framework cannot be easily extended to generation tasks.
3. The human evaluation to me is not that reliable. There are 18 annotators but with only 230 instances. I am concerned that there can be a bid variance between different annotators. Also, I did not find the human annotation agreement. Could please the authors remind me if I am wrong?

[1] See What LLMs Cannot Answer: A Self-Challenge Framework for Uncovering LLM Weaknesses

**Questions:**

Please see above.

---

> ### Author Response · Authors · 2024-11-24
> **Rebuttal by Authors**
>
> Thank you for your valuable review and criticism. Please find our comments below.
>
> > Note: We've marked "EuP9" next to relevant changes in the general comments to make it easier for you to navigate.
>
> **W1. Other works focusing on qualitative evaluation**
>
> Thank you for bringing [1] to our attention. The Self-Challenge framework emphasizes exposing model failure patterns and generating an evaluation dataset that poses challenges for frontier models. Rather than focus on creating a challenging dataset that can be used to identify failure modes, our focus is on fine-grained qualitative evaluations that can be used to compare and contrast the strengths of different models—-we are not aware of other works that propose to do this in an automatic way. Indeed, the evaluation set produced via Self-Challenge (which is evaluated quantitatively) might be used to create complementary Report Cards that summarize relative differences between models. We leave this for future work, but will include the citation to [1] in our paper. We also note that a large part of our contribution is our approach to evaluating the qualitative evaluation (i.e., the 3 metrics that assess specificity, faithfulness, and interpretability of qualitative evaluations).
>
> **W2. Limited task selection; cannot be easily extended to generation tasks**
>
> Generation tasks where LLMs produce extended and open-ended are hard to evaluate. However, we’ve made progress toward such open-ended tasks with no ground truth answer. The Chinese grammar correction dataset (Appendix Section B.5) contains a suggested correction for each erroneous sentence, but there usually are multiple ways to do the corrections. In our experiments, we chose to exclude the suggested corrections from both the PRESS algorithm and Report Card assessment tasks like the contrastive tasks. Therefore, this task is open-ended.  Our results in Figure 7 validate show that Report Cards maintain high faithfulness even for such open-ended generation tasks.
>
> Since Report Cards are designed to evaluate specific model capabilities against some criteria (or sub-topics), evaluating completely open-ended generation tasks is a significant challenge without an established rubric. This is why we limited our scope to tasks that maintain some degree of structured evaluation criteria, such as the Chinese Grammar correction task, rather than including purely open-ended generation scenarios.
>
> **W3. unreliable human evaluation**
>
> As shown in the Appendix Section E, volunteers need to read the question, model’s answer, and an excerpt of a Report Card for each question, which is time-consuming and difficult to administer at scale. For this reason the collection process is rather slow, and we had to choose between collecting labels for more questions or collecting duplicate labels in order to compute human agreement. We agree that human agreement would be an asset here, and are in the process of collecting more human annotations. As of this response we have collected 100 more labels (330 in total now), and we are confident that we will be able to collect sufficient annotations to compute statistically meaningful agreement statistics.

---

> > ### Comment · Reviewer_EuP9 · 2024-11-26
> >
> > Thank the authors for their reply.
> >
> > 1. I do not think Self-challenge is quantitatively-based. The summarized descriptions (of LLM failures) can be regarded as qualitative analyses.
> > 2. Thanks for the clarification of "task limitation". Do the authors think it is possible to apply their method to LLM-based evaluators, such as G-eval? That is, for example, given a text summarization task, the proposed method generates more fine-grained natural language criteria, which can be used for G-eval?
> > 3. Sorry that I did not completely follow the authors' response that "we will be able to collect sufficient annotations to compute statistically meaningful agreement statistics." Can they show me the results?

---

> > > ### Author Response · Authors · 2024-12-01
> > > **Clarifications on Self-Challenge, Report Cards, and Human Agreement**
> > >
> > > Thank you for your follow-up comments.
> > >
> > > 1. We appreciate your correction regarding Self-Challenge. To clarify, we didn't claim that Self-Challenge is quantitatively-based; we agree that its pattern descriptions constitute qualitative analyses. While Pattern Optimization in Self-Challenge generates optimized descriptions of LLM failure patterns, our objectives differ. We aim to **automatically** produce comprehensive interpretable evaluations covering aspects of LLM behavior - strengths, general behaviors, and limitations - while also providing multiple metrics to *automatically assess these evaluations*. Thus, although Self-Challenge's pattern descriptions share similar ideas with Report Cards in terms of qualitative evaluation, the purpose and methodology remain distinct.
> > > 2. Yes, thank you for this insightful suggestion. Report Cards are naturally structured with orthogonal criteria (sub-topics), as demonstrated in our supplementary materials. The PRESS algorithm begins with pre-defined criteria based on the topic, and PRESS progressively adds more criteria while generating Report Cards. For a task like text summarization, we could generate a Report Card based on a dataset of human demonstrations, describing human behavior for each criterion. Each Report Card entry would then naturally serve as a detailed and concrete criterion for LLM-based evaluators like G-eval to score against.
> > > 3. We apologize for the lack of clarity in our previous response regarding human agreement statistics. As explained in our previous response, the data collection process is particularly challenging and it requires participants with STEM backgrounds. Our previous data and the newly collected 100+ entries correspond to different Report Cards and queries, preventing us from computing meaningful human agreement statistics. However, we are actively working to gather more resources and collect sufficient human labels to analyze both human-LLM alignment and human agreement.

---

> > > > ### Comment · Reviewer_EuP9 · 2024-12-01
> > > >
> > > > Let me be very clear: does the phrase "we are actively working..." refer to the fact that the authors have been collecting data for alignment and agreement analyses, or simply refer to a promise that you will collect data?
> > > >
> > > > If it is the former case, when will the results be expected to be complete?
> > > > Could the authors show it before the deadline?

---

> > > > > ### Author Response · Authors · 2024-12-03
> > > > > **Data Collection Update**
> > > > >
> > > > > Unfortunately we were unable to get sufficient data for agreement analyses by the response deadline, but we expect to be able to do so over the course of the next month.

---

### Author Response · Authors · 2024-11-24
**Summary of Paper Revision and General Comments**

We thank all reviewers for their constructive feedback and patience , and we have responded to each reviewer individually. We have also uploaded a **Paper Revision** including additional results and clarifications. Modifications made to the paper are highlighted as $\textrm{\color{blue}blue}$.

- $\textrm{\color{blue}Section 2.2}$ (page 3): We changed the Ground-truth Elo to Oracle Elo. (reviewer D9R8)
- $\textrm{\color{blue}Section 3.2}$ (page 6): We further clarify the notation for the few-shot baseline. (reviewer BghS)
- $\textrm{\color{blue}Section 4}$ (page 9 and 10): We added the reference to [1]. (reviewer EuP9)
- $\textrm{\color{blue}Appendix Section C.2}$ (page 18 and 20): We added the ablation study of different card writer. (reviewer f1vs)
- $\textrm{\color{blue}Appendix Section D}$ (page 20): We updated the implementation details for generating and assessing Report Cards on the Chinese Grammar Correction dataset. (reviewer EuP9 and BghS)

We also attached the following files to the supplementary material (as the `supplementary material.zip`):

- `report_cards.html` contains an HTML file that allows for comparison of Report Cards across different models. We mapped question-completion pairs from the training sets to their corresponding descriptions after generation. This augmentation helps us to better understand model behavior.
- `Report Card for Claude 3.5 Sonnet on MATH Algebra.pdf` contains a complete Report Card for Claude 3.5 Sonnet on the MATH Algebra topic.
- `Report Card for Llama-3-8b on Adv. AI Safety Self Awareness of AI.pdf` contains a complete Report Card for Llama 3 8b on the Advanced AI Safety dataset with the topic Self Awareness of AI.

[1] See What LLMs Cannot Answer: A Self-Challenge Framework for Uncovering LLM Weaknesses

---

### Author Response · Authors · 2024-11-26
**Rebuttal Revision Reminder**

Dear Reviewers,

We appreciate the time and effort you have invested in evaluating our work. As a gentle reminder, the revision period is scheduled to end on Nov 26th (Anywhere on Earth).

In our rebuttal, we have strived to address all concerns and questions raised. However, if there are any additional questions or points that require further clarification, we would be happy to provide further clarification before the deadline.

Respectfully,
Authors of "Report Cards: Qualitative Evaluations"

---

### Meta-Review · Area_Chair_eGbH · 2024-12-23

**Metareview:**

The paper introduces "Report Cards" as a qualitative evaluation and description of large language models through human-interpretable natural language summaries. The authors proposed methods to evaluate the quality of generated cards. They also proposed an iterative algorithm called PRESS for generating these summaries. It basically allow the generating model to improving its summary by asking candidate LLMs quizs in multiple iterations. Experimental results show that Report Cards provide fine-grained insights into LLM capabilities.

Strengths
1. The idea of having qualitative evaluation instead of quantitive evluation of LLMs is interesting, and it is becoming useful with the spreading of LLM's practical usages.
2.  The introduction of specificity, faithfulness, and interpretability as evaluation metrics of generated cards provides a structured framework of evaluating the quality of Report Cards.
3. The method reveals nuanced capabilities of LLMs, such as reasoning and contextual understanding, that are challenging to assess with traditional benchmarks.

Weaknesses
1. The experiments focus on specific datasets and tasks, limiting generalizability to broader or more open-ended domains
2. The paper relies on limited human evaluations, raising concerns about the reliability of interpretability assessments.
3. Several reviewers noted a lack of clarity on how Report Cards provide actionable insights, particularly in practical applications deployment of LLMs.

While the paper presents a compelling and novel approach to evaluating LLMs qualitatively, it falls short in addressing key issues of task generalizability, human evaluation reliability, and its potential in practical applications. The reviewers are divided, with one leaning towards acceptance due to the paper’s novel contribution, while others express reservations regarding reliability and insights. The rebuttal effectively addressed some concerns, but critical gaps remain in human evaluation statistics and broader applicability.

**Additional Comments On Reviewer Discussion:**

The potential of usage of the Report Cards was questioned. The reviewer was skeptical about whether actionable insights or practical use could be derived from the study. The authors clarified that the primary intent was to provide interpretability and qualitative insights into LLM performance, rather than immediate actionable insights. I believe the response partially addressed the concerns, but the lack of actionable insights remained a major limitation of this work.

Reviewer questioned that the paper lacks open-ended generation tasks. The authors highlighted progress in handling semi-open-ended tasks, such as the Chinese Grammar Correction dataset, and argued that extending to fully open-ended tasks would require additional rubrics. I believe the authors’ rebuttal was good enough given the stated scope of the paper.

---

### Decision · Program_Chairs · 2025-01-22

Reject